# Relationship between Psychological Status and Health Behaviors during the Coronavirus Disease Pandemic in Japanese Community-Dwelling Older Adults

**DOI:** 10.3390/ijerph182111512

**Published:** 2021-11-02

**Authors:** Shuichi Hara, Hiroko Miura, Tsuyoshi Hita, Sahara Sasaki, Hidetoshi Ito, Yumi Kozaki, Yoshiko Kawasaki

**Affiliations:** 1School of Clinical Psychology, Kyushu University of Health and Welfare, 1714-1 Yoshino-machi, Nobeoka, Miyazaki 882-8508, Japan; 2Division of Disease Control and Epidemiology, School of Dentistry, Health Sciences University of Hokkaido, 1757 Ishikari-Tobetsu, Hokkaido 061-0293, Japan; hmiura@hoku-iryo-u.ac.jp; 3School of Social Welfare, Kyushu University of Health and Welfare, 1714-1 Yoshino-machi, Nobeoka, Miyazaki 882-8508, Japan; t.hita@phoenix.ac.jp (T.H.); s-sasaki@phoenix.ac.jp (S.S.); ito-hide@phoenix.ac.jp (H.I.); yumipon-10th.100p@outlook.jp (Y.K.); kawasaki@phoenix.ac.jp (Y.K.)

**Keywords:** COVID-19, older adults, health behavior, anxiety, mental stress

## Abstract

The coronavirus disease (COVID-19) continues to be a widespread pandemic. We investigated the relationship between anxiety/stress and health behaviors during the COVID-19 pandemic in homebound Japanese older adults during January and February 2021. We surveyed 1507 community-dwelling, older Japanese adults using a self-administered questionnaire on primary attributes, including family structure, evaluation of psychological anxiety/stress, and health behaviors. Participants were divided into four anxiety/stress groups based on the frequency of experiencing anxiety/stress, and their association with health behaviors was analyzed using bivariate and multivariate analyses. Responses were received from 469 (31.1%) respondents. In the bivariate analysis, age and family structure were significantly associated with anxiety/stress (*p* < 0.01). The health behaviors significantly associated with anxiety/stress were walking, balanced eating habits, limited snacking, regular lifestyle, and dental visits. Logistic regression analysis was performed using the variables in the bivariate analysis that showed a significant association with anxiety/stress status as independent variables. Finally, age and dietary habits were significantly associated with anxiety/stress status. No significant associations were found between any other variables. Among older adults living in the rural areas of Japan during the COVID-19 pandemic, anxiety/stress status was significantly associated with age and dietary habits but not with other health behaviors.

## 1. Introduction

Since the first case of coronavirus disease (COVID-19) was confirmed in Japan in January 2020, more than 1.5 million cases of infection have been reported nationwide as of early September 2021 [1]. Furthermore, in April 2020, a state of emergency was declared in Japan, which led to requests for cooperation, such as refraining from going out of the house unnecessarily and restricting events and the use of public facilities [2].

Since it was reported that the mortality rate of older adults infected with COVID-19 was high, the system for providing nursing care for the elderly in Japan was also severely restricted. A survey by one of the long-term care insurance services in Japan of homebound older adults using day-care services reported that >70% of the older adults using day-care services themselves and their families refrained from using the services, showing a decline in activities of daily living, cognitive function, and physical activity [3]. As for the older adults living in the community, restrictions were imposed on going out, the implementation of care prevention classes organized by municipalities was postponed, and the implementation of day-care services by long-term care insurance providers was restricted. Several studies have shown that inactivity in daily life among the elderly increases the risk of frailty and muscle wasting due to loss of muscle strength and reduction in physical fitness, motor function, nutritional status, and opportunities for interaction and communication [4,5,6]. In addition, the risk of increased anxiety/stress caused by decreased communication with others also requires sufficient consideration [7]. The psychological burden in older individuals may include stress caused by external factors and anxiety, which are psychological factors resulting from stress [8]. Several studies have reported on the factors that contribute to the psychological burden on older adults in the context of the COVID-19 pandemic [9,10]. Findings [9] have reported that the prevalence of anxiety disorders during the COVID-19 pandemic occurred in the United States was significantly higher than the prevalence reported after other significant traumatic events. Moreover, people who experience social and emotional stressors, especially stress due to economic problems and loneliness, are more likely to have post-traumatic stress disorder (PTSD) and anxiety disorders. On the other hand, the findings in Canada [10] reported that subjects with high COVID-19 anxiety tended to be socially distant and spent more time in sedentary activities such as watching television. These findings suggest that in the spread of COVID-19 infection, a balance between physical activity promotion and mental health measures should be provided. However, it is necessary to accumulate research findings by each region, as people are affected by the prevalence of COVID-19 in different ways and based on their area of residence and the human resources and public services that support the older adults in that particular area.

During the COVID-19 pandemic, continuous preventive activities such as avoiding dense, close, and hermetic contact; understanding and implementing appropriate lifestyle habits, such as exercise and healthy diet, according to each individual’s health condition; and measuring body temperature every morning are required [11]. In addition, it is essential to maintain healthy behaviors in the daily lives of the elderly without increasing their psychological burden. Several studies have reported on the health behavior of older adults at home during the COVID-19 pandemic [12,13]. However, few studies have directly analyzed the relationship between states of anxiety/stress in community-dwelling older adults and their health behaviors. Thus, the purpose of this study was to clarify the relationship between anxiety and mental stress and the implementation of health behaviors during the COVID-19 pandemic among Japanese community-dwelling older adults.

## 2. Participants and Methods

### 2.1. Participants and Research Design

A total of 1507 community-dwelling older individuals living in Kijo, Miyazaki Prefecture, in the Kyushu area of Japan, were included in the study. A self-administered questionnaire on anxiety, stress, and health behaviors was sent to the participants through the local government and was collected after a 3-week retention period. The questionnaire was anonymous and did not contain any personal identifiable information. As a reminder to return the questionnaires, a call for their return was made via town hall announcements. The number of respondents was 469, and the collection rate was 31.1%. This study was designed as a cross-sectional survey using only the results of the questionnaires.

### 2.2. Survey Items

The primary attributes included sex, age, occupation, and family structure. The survey items included anxiety/stress and health behaviors that had been implemented before the spread of the COVID-19 infection and the current situation. We also asked the participants about their health behaviors during the COVID-19 pandemic situation and before the spread of COVID-19, which included their efforts related to exercise (walking, jogging, use of sports clubs, participation in exercise clubs and classes, ground golf, and light exercises) and daily life (consideration of moving method, balanced eating habits, limitation of snacks, limitation of drinking and smoking, regular lifestyle, and dental visits). We classified the health behaviors into four groups as follows: not practiced before the spread of COVID-19 infection and at the time of the survey (not practiced vs. not practiced group), practiced before the spread of infection, but stopped at the time of the survey (practiced vs. not practiced group), not practiced before the spread of infection, but practiced at the time of the survey (not practiced vs. practiced group), and continued both before the spread of infection and at the time of the survey (practiced vs. practiced group). The assessment of anxiety/stress was based on the current situation. The presence or absence of anxiety about their present and future life was examined using a two-category scale. First, the stress status was assessed by the number of items that corresponded to nine typical stress items, such as infection, inability to engage in hobbies, discord in the family, relations with neighbors, difficulty in interacting with friends, difficulty with household chores, financial problems, and unemployment [14]. Second, the stressed group was defined as those with three or more items indicating the presence of a moderate or more significant stress factor based on the previous research [9] and average items and one standard deviation (SD). Based on these evaluation results, the participants were divided into four groups: no anxiety/stress group, mild anxiety/stress group, moderate anxiety/stress group, and severe anxiety/stress group (Table 1).

### 2.3. Statistical Analysis

To adjust for confounding factors, bivariate and multivariate analyses were conducted using a phased combination. First, a chi-squared test was used for bivariate analysis to examine the association between primary attributes and each health behavior group among the four anxiety/stress groups. Next, a logistic regression analysis was conducted in which the dependent variable was the anxiety/stress state, and the independent variables were those variables that had significant associations in the bivariate analysis. The dependent variable was set to 0 for the no anxiety/stress group and 1 for the mild-to-severe anxiety/stress group. Health behaviors selected as independent variables were analyzed as 1 for the no-practice and practice–practice groups and 0 for the no-practice and practice–no-practice groups. Age was based on the average life expectancy of Japanese people in 2019 (81.41 years for men and 87.45 years for women) and was analyzed as 1 for age <80 years and 0 for age ≥80 years. Households with four or more members were grouped together, and then the number of households was entered. The statistical analysis package SPSS Statistics Japanese version 27.0 (IBM) was used for the analysis, and statistically significant differences were considered to exist when the *p*-value was less than 0.05 in a two-tailed test.

## 3. Results

### 3.1. Respondents’ Attributes

The distribution of the primary attributes of the respondents is shown in Table 2. Among the respondents, 207 (44.1%) were men and 249 (53.1%) were women. The most common age group was 70–74 years (127 respondents, 27.1%). Regarding occupation, 272 (58.0%) were unemployed, and 96 (20.5%) were self-employed. In the two-person households, 189 (75.6%) were living with a spouse.

Table 3 shows the status of health behaviors at the time of the survey. The behaviors that were implemented continuously and at a high rate from before the spread of the infection to the time of the survey were walking (135 respondents, 28.8%), balanced eating habits (185 respondents, 39.4%), and regular lifestyle (131 respondents, 27.9%).

### 3.2. Anxiety and Mental Stress

In the survey, 309 respondents (65.9%) felt anxious, of whom, 70 respondents (15.0%) also felt anxious about the future. Table 4 lists the items related to mental stress. A total of 191 respondents (41.4%) were worried about COVID-19; 140 (29.9%) were concerned about their health. The mean and standard deviation (SD) of stress items were 1.4 (SD 1.6; minimum–maximum items: 0–9), and 91 (19.4%) felt stressed by three or more items.

### 3.3. Relationship between Psychological Status and Primary Attributes or Health Behaviors

Table 5 and Table 6 show the bivariate association between anxiety/stress status, primary attributes, and exercise habits. By age group, the number of participants in the high anxiety/stress group was 27 (27.3%) in the 65–69 years age group and 24 (26.1%) in the 75–79 years age group, which was significantly higher than that in the ≥80 years age group (*p* < 0.01). In terms of family structure, the number of participants in the severe anxiety/stress group was 56 (22.4%) in the two-person households and 15 (26.8%) in households with four or more members (*p* < 0.01). Regarding exercise habits, there was a significant difference in the anxiety/stress groups only with regard to walking (*p* < 0.05; Table 6).

Table 7 shows the relationship between anxiety/stress status and daily life. There were significant differences between the anxiety/stress groups in four items: balanced eating habits, limited snacking, regular lifestyle, and dental visits. For example, the 57 (30.8) participants in the severe anxiety/stress group practiced balanced eating habits most frequently and continuously.

Table 8 shows the results of logistic regression analysis. In terms of age, anxiety/stress was 2.24 times (95% CI, 1.42–3.54) significantly higher (*p* < 0.01) in the <80 years age group than in the >80 years group. In regard to health behaviors, anxiety/stress was significantly higher in those who practiced balanced eating habits, i.e., 2.03 times (95% CI, 1.15–3.60) higher than in those who did not practice balanced eating habits (including those who discontinued) (*p* < 0.05).

## 4. Discussion

The results of the present study indicate that approximately 70% of the respondents felt anxious during the COVID-19 pandemic and that a high percentage of the stress they experienced was related to infectious diseases and health. In addition, the results of this study revealed a significant relationship between anxiety/stress status, age, and balanced eating habits. In the bivariate analysis, the number of household members, limited snacking, dental visits, and regular lifestyle were significantly associated, but no significant association was found after adjusting for confounding factors by logistic regression analysis.

Among the participants in this study, the number of those in the high anxiety/stress group was significantly higher in the <80 years age group. A similar trend has been suggested by research findings on health-related quality of life in Japanese community-dwelling older adults [15]. The health-related QOL sub-items that showed changes before and after the COVID-19 pandemic among individuals in their 60s and 70s were “physical functioning”, “social functioning”, and “mental health”, whereas no significant decrease was observed in the age group of 80 years in the previous study. These results suggest that those under 80 years of age who interact with others may be more susceptible to the effects of social changes caused by the COVID-19 pandemic. Conversely, other countries have reported that positive lifestyles, such as sleep and regular mealtimes, healthy diet, cognitive stimulation, and exercise, as well as quick and straightforward provision of reliable information, can reduce anxiety among the elderly during a pandemic and have positive effects on mental health [10,12]. Thus, the results of this study indicate that measures such as raising awareness and providing information to prevent infection are incredibly significant in reducing anxiety caused by COVID-19 among older adults in their 60s and 70s who have relatively preserved activities of daily living.

Balanced dietary habits are the most common health behavior. Simpo et al. [16], in their study on changes in lifestyle and dietary habits during the novel COVID-19 infection epidemic in Japanese adults aged 20–64 years, found that healthy dietary habits were associated with anxiety about COVID-19, changes in exercise and sleep time, and smoking, which approximated the findings of the present study. Low nutrition in old age is a significant cause of frailty and sarcopenia, and older adults need to pay more attention to their dietary habits than younger people to maintain a healthy life. As COVID-19 infection control measures impose restrictions on going out, an expansion of food delivery services is strongly needed to support the dietary needs of the elderly. In addition, the results of this study suggest that the establishment of healthy eating habits among older adults may become a normal coping behavior not only for their physical health but also for dealing with their anxiety/stress.

Dental visits are a specific care need constrained by COVID-19 prevalence in Japan [17,18]. This study also showed that 66 people had foregone dental visits. Bivariate analysis showed a significant association between anxiety/stress and dental visits, while logistic regression analysis showed no significant association after adjusting for confounders. The reduced dental visits among the elderly may have been influenced by the fact that routine dental services were partially restricted when COVID-19 infection first began to spread, rather than by anxiety and stress factors. In the elderly prone to periodontal complications, interruption of home life and dental care due to COVID-19 may exacerbate periodontal disease [19]. Weber et al. [20] reported that the decrease in dental-related quality of life of older adults requiring care during the COVID-19 pandemic was significantly associated with decreased social support. Further studies are needed to determine the impact of necessary social support during the spread of COVID-19 infection on dental visits in Japan.

One of the limitations of this study is that the collection rate of the questionnaire was 31.3%, which may not adequately capture the situation of the entire surveyed older population. This limitation could be because it was difficult for older adults with reduced activities of daily living to respond to the questionnaire, especially considering that it was a self-administered questionnaire. In addition, the evaluation of factors related to anxiety/stress was based on a limited number of survey items, which have further methodological limitations. In addition, since the study area was rural, it was not possible to understand the situation of older adults living in urban areas.

Despite these limitations, few studies have reported the association between psychological stress and health behaviors in older adults during the spread of COVID-19; this is the first study to report such findings in Japan. Furthermore, poor dietary quality can easily lead to low nutritional status. Therefore, it is necessary to continue improving the environment to support the dietary habits of older adults. In the future, similar studies should be conducted on older urban adults.

## 5. Conclusions

In conclusion, we investigated the association between anxiety/stress and health behaviors during the spread of COVID-19 in Japanese homebound older adults using a combination of bivariate and multivariate analyses. We found that age and dietary habits were significantly associated with anxiety and stress.

## Figures and Tables

**Table 1 ijerph-18-11512-t001:** Selection criteria for anxiety and stress groups.

Group	Anxiety (Present)	Anxiety (Future)	Number of Stressors
No anxiety/stress	none	none	<3
Mild anxiety/stress	∘ or ×	∘ or ×	<3 or ≥3
Moderate anxiety/stress	∘ or ×	∘ or ×	<3 or ≥3
Severe anxiety/stress	∘	∘	≥3

Mild anxiety/stress group: One of the three conditions applies. Moderate anxiety/stress group: Two of the three conditions applies.

**Table 2 ijerph-18-11512-t002:** Characteristics of the 469 respondents.

Sex	Male	207 (44.1)
	Female	249 (53.1)
	Others	13 (2.8)
Age	65−69 years	99 (21.1)
	70−74 years	127 (27.1)
	75−79 years	92 (19.6)
	80−84 years	85 (18.1)
	>85 years	62 (13.2)
	unknown	4 (0.9)
Work	Unemployed	272 (58.0)
	Independent business	96 (20.5)
	Company employee	19 (4.1)
	Part-timer	32 (6.8)
	Others	28 (6.0)
	Unknown	22 (4.7)
Household composition	One	81 (17.3)
	Two	250 (53.3)
	Three	80 (17.1)
	>Four	56 (11.9)
	Unknown	2 (0.4)

Numbers of respondents (%).

**Table 3 ijerph-18-11512-t003:** Status of health behavior at the time of the survey (multiple responses).

	Not Practiced ^1^	Practiced–Not Practiced ^2^	Not Practiced–Practiced ^3^	Practiced–Practiced ^4^
Walking	250 (53.3)	84 (17.9)	0 (0.0)	135 (28.8)
Jogging	440 (93.8)	14 (3.0)	0 (0.0)	15 (3.2)
Use of sports clubs	435 (92.8)	30 (6.4)	0 (0.0)	4 (0.9)
Participation in classes	397 (84.6)	65 (13.9)	0 (0.0)	7 (1.5)
Ground golf	420 (89.6)	40 (8.5)	0 (0.0)	9 (1.9)
Calisthenics	374 (79.7)	46 (9.8)	3 (0.6)	46 (9.8)
Ingenious mobility	343 (73.1)	49 (10.4)	1 (0.2)	76 (16.2)
Balanced eating habits	245 (52.2)	34 (7.2)	5 (1.1)	185 (39.4)
Limited snacking	339 (72.3)	45 (9.6)	8 (1.7)	77(16.4)
Limited drinking	373 (79.5)	45 (9.6)	0 (0.0)	51 (10.9)
Limited smoking	452 (96.4)	8 (1.7)	1 (0.2)	8 (1.7)
Regular lifestyle	297 (63.3)	37 (7.9)	4 (0.9)	131(27.9)
Dental visits	292 (62.3)	66 (14.1)	1 (0.2)	110 (23.5)

Multiple responses. Number of respondents (%). ^1^ Not practiced before the spread of COVID-19 infection and at the time of the survey. ^2^ Practiced before the spread of infection but stopped at the time of the survey. ^3^ Not practiced before the spread of infection but practiced at the time of the survey. ^4^ Continued both before the spread of infection and at the time of the survey.

**Table 4 ijerph-18-11512-t004:** “Stressful” contents (multiple responses).

Contents	Number
Infection	191
Own health/chronic illness/family health condition	140
Inability to engage in hobbies	56
Discord in family	49
Relationships with neighbors	37
Difficulty in interacting with friends	31
Difficulty with household chores	30
Financial problem	28
Unemployment or job loss	24

**Table 5 ijerph-18-11512-t005:** Age, sex, employment, and family structure during the COVID-19 pandemic in respondent groups with different levels of anxiety and stress.

	No Anxiety and Stress *n* = 108	Mild Anxiety and Stress *n* = 147	Moderate Anxiety and Stress *n* = 117	Severe Anxiety and Stress *n* = 97	*p*-Value
Age, years: 65–69	13 (13.1)	27 (27.3)	32 (32.3)	27 (27.3)	**
70–74	23 (18.1)	46 (36.2)	35 (27.6)	23 (18.1)	
75–79	20 (21.7)	29 (31.5)	19 (20.7)	24 (26.1)	
80–84	33(38.8)	20 (23.5)	19 (22.4)	13 (15.3)	
Over 85	17(27.4)	24 (38.7)	11 (17,7)	10 (16.1)	
unknown	2 (50.0)	1 (25.0)	1(25.0)	0 (0,0)	
Sex, Male	58 (28.0)	61 (29.5)	52 (25.1)	36 (17.4)	ns
Female	45 (18.1)	84 (33.7)	61 (24.5)	59 (23.7)	
Others	5 (38.5)	2 (15.4)	4 (30.8)	2(15.4)	
Work, Unemployed	71(24.1)	94 (32.0)	64(21.8)	65 (22.1)	ns
Employed	37 (21.1)	53(30.3)	53(30.3)	32 (18.3)	
Family structure, Single	16 (19.8)	31 (38.3)	21 (25.9)	13 (16.0)	**
Two	56 (22.4)	71 (28.4)	67 (26.8)	56 (22.4)	
Three	19 (23.8)	31 (38.8)	17 (21.3)	13 (16.3)	
>Four	15 (26.8)	14 (25.0)	12 (21.4)	15 (26.8)	

Number of respondents (%). ** *p* < 0.01. ns: not significant.

**Table 6 ijerph-18-11512-t006:** Exercise during the COVID-19 pandemic in respondent groups with different levels of anxiety and stress.

	No Anxiety and Stress *n* = 108	Mild Anxiety and Stress *n* = 147	Moderate Anxiety and Stress *n* = 117	Severe Anxiety and Stress *n* = 97	*p*-Value
**Walking**					*
Not practiced	68 (27.2)	83 (33.2)	60 (24.0)	39 (15.6)	
Practiced–not practiced	10 (11.9)	27 (32.1)	25 (29.8)	22 (26.2)	
Not practiced–practiced	0 (0.0)	0 (0.0)	0 (0.0)	0 (0.0)	
Practiced–practiced	30 (22.2)	37(27.4)	32 (23.7)	36 (26.7)	
**Jogging**					ns
Not practiced	102 (23.2)	137 (31.1)	107 (24.3)	94 (21.4)	
Practiced–not practiced	2 (14.3)	5 (35.7)	5 (35.7)	2 (14.3)	
Not practiced–practiced	0(0.0)	0 (0.0)	0 (0.0)	0 (0.0)	
Practiced–practiced	4 (26.7)	5 (33.3)	5 (33.3)	1 (6.7)	
**Use of sports clubs**					ns
Not practiced	105 (24.1)	135 (31.0)	106 (24.4)	89 (20.5)	
Practiced–not practiced	3 (10.0)	10 (33.3)	10 (33.3)	7 (23.3)	
Not practiced–practiced	0 (0.0)	0 (0.0)	0 (0.0)	0 (0.0)	
Practiced–practiced	0 (0.0)	2 (50.0)	1 (25.0)	1 (25.0)	
**Participation in classes**					ns
Not practiced	96 (24.2)	129 (32.5)	98 (24.7)	74 (18.6)	
Practiced–not practiced	10 (15.4)	15 (23.1)	18 (27.7)	22 (33.8)	
Not practiced–practiced	0 (0.0)	0 (0.0)	0 (0.0)	0 (0.0)	
Practiced–practiced	2 (28.6)	3 (42.9)	1 (14.3)	1 (14.3)	
**Ground golf**					ns
Not practiced	104 (24.8)	133 (31.7)	102 (24.3)	81 (19.3)	
Practiced–not practiced	2 (5.0)	11 (27.5)	13 (32.5)	14 (35.0)	
Not practiced–practiced	0 (0.0)	0 (0.0)	0 (0.0)	0 (0.0)	
Practiced–practiced	2 (22.2)	3 (33.3)	2(22.2)	2 (22.2)	
**Calisthenics**					ns
Not practiced	90 (24.1)	121(32.4)	92 (24.6)	71 (19.0)	
Practiced–not practiced	8 (17.4)	11 (23.9)	11 (23.9)	16 34.8)	
Not practiced–practiced	0 (0.0)	1 (33.3)	2 (66.7)	0 (0.0)	
Practiced–practiced	10 (21.7)	14 (30.4)	12 (26.1)	10 (21.7)	

Number of respondents (%). * *p* < 0.05. ns: not significant.

**Table 7 ijerph-18-11512-t007:** Lifestyle of respondent groups with different levels of anxiety and stress during the COVID-19 pandemic.

	No Anxiety and Stress *n* = 108	Mild Anxiety and Stress *n* = 147	Moderate Anxiety and Stress *n* = 117	Severe Anxiety and Stress *n* = 97	*p*-Value
**Consideration of moving method ^1^**					ns
Not practiced	83 (24.2)	108 (31.5)	83 (24.2)	69 (20.1)	
Practiced–not practiced	15 (30.6)	13 (26.5)	10 (20.4)	11 (22.4)	
Not practiced–practiced	0 (0.0)	0 (0.0)	1 (100.0)	0 (0.0)	
Practiced–practiced	10 (13.2)	26 (34.2)	23 (30.3)	17 (22.4)	
**Balanced eating habits**					***
Not practiced	76 (31.0)	82 (33.5)	57 (23.3)	30 (12.2)	
Practiced–not practiced	5 (14.7)	10 (29.4)	9 (26.5)	10 (29.4)	
Not practiced–practiced	2 (40.0)	1 (20.0)	2 (40.0)	0 (0.0)	
Practiced–practiced	25 (13.5)	54 (29.2)	49 (26.5)	57 (30.8)	
**Limitation of snacking**					**
Not practiced	87 (25.7)	114 (33.6)	89 (26.3)	49 (14.5)	
Practiced–not practiced	8 (17.8)	11 (24.4)	11 (24.4)	15 (33.3)	
Not practiced–practiced	2 (25.0)	2 (25.0)	1 (12.5)	3 (37.5)	
Practiced–practiced	11(14.3)	20 (26.0)	16 (20.8)	30 (39.0)	
**Limitation of drinking**					ns
Not practiced	94 (25.2)	109 (29.2)	97 (26.0)	73 (19.6)	
Practiced–not practiced	6 (13.3)	16 (35.6)	10 (22.2)	13 (28.9)	
Not practiced–practiced	0 (0.0)	0 (0.0)	0 (0.0)	0 (0.0)	
Practiced–practiced	8 (15.7)	22 (43.1)	10 (19.6)	11 (21.6)	
**Limitation of smoking**					ns
Not practiced	108 (23.9)	141 (31.2)	111 (24.6)	92 (20.4)	
Practiced–not practiced	0 (0.0)	2 (25.0)	2 (25.0)	4 (50.0)	
Not practiced–practiced	0 (0.0)	1 (100.0)	0 (0.0)	0 (0.0)	
Practiced–practiced	0 (0.0)	3 (37.5)	4 (50.0)	1 (12.5)	
**Regular lifestyle ^2^**					***
Not practiced	82 (27.6)	99 (33.3)	77 (25.9)	39 (13.1)	
Practiced–not practiced	5 (13.5)	10 (27.0)	3 (8.1)	19 (51.4)	
Not practiced–practiced	0 (0.0)	0 (0.0)	3 (75.0)	1 (25.0)	
Practiced–practiced	21 (16.0)	38 (29.0)	34 (26.0)	38 (29.0)	
**Dental visits**					*
Not practiced	83 (28.4)	94 (32.2)	67 (22.9)	48 (16.4)	
Practiced–not practiced	9 (13.6)	17 (25.8)	17 (25.8)	23 (34.9)	
Not practiced–practiced	0 (0.0)	0 (0.0)	1 (100.0)	0 (0.0)	
Practiced–practiced	16 (14.5)	36 (32.7)	32 (29.1)	26 (23.6)	

Number of respondents (%). ^1^ Walk as much as possible, use a bicycle for transportation, etc. ^2^ Going to bed early, getting up early, keeping a regular rhythm in life, etc. * *p* < 0.05, ** *p* < 0.01, *** *p* < 0.001. ns: not significant.

**Table 8 ijerph-18-11512-t008:** Logistic regression analysis of anxiety and stress as dependent variables.

Independent Variable	B	Standard Error	Wald	*p*-Value	Odds Ratio	95% CI		
**Walking,** practicing = 1, no habit = 0	−0.296	0.266	10.234	0.267	0.744	0.441	-	1.254
**Balanced eating habits****,** practicing = 1, no habit = 0	0.709	0.292	5.876	0.015	2.032	1.145	-	3.604
**Limit snacking,** practicing = 1 no habit = 0	0.042	0.369	0.013	0.909	1.043	0.506	-	2.148
**Regular lifestyle,** practicing = 1, no habit = 0	0.234	0.318	0.540	0.463	1.264	0.677	-	2.358
**Dental visits,** practicing = 1, no habit = 0	0.368	0.323	1.300	0.254	1.444	0.768	-	2.718
**Age**, <80 y = 1 >80 y = 0	0.806	0.234	11.861	0.001	2.239	1.415	-	3.541
**Family structure**, +1	−0.072	0.128	0.316	0.574	0.930	0.723	-	1.197
**Constant**	0.577	0.359	2.588	0.108	1.781			

## Data Availability

Not applicable.

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
