# Peer review of "Relationship between Psychological Status and Health Behaviors during the Coronavirus Disease Pandemic in Japanese Community-Dwelling Older Adults"

_ijerph, 2021, doi:10.3390/ijerph182111512_

Round 1
Reviewer 1 Report
Dear Authors
In general, the document is clear, comprehensible and expresses in a concise way the necessary and sufficient information to communicate the results derived from the investigation. Here are some brief comments on each section of the manuscript.
Introduction
The section adequately communicates the background and reasons for carrying out the study, however, it could be deepened and even mention what are the factors that can contribute to increasing stress levels in older adults, for example, what were the findings in the studies cited (US and Canada)? Since this evidence could be used to contrast the results of this research. Outside of this detail, I consider the rest of the introduction to be adequate.
Methods
A clear description of the study design and research methods is made, and the type of statistical analysis that was implemented is clearly defined. However, the manuscript would be enriched if the authors add the information related to the anxiety / stress questionnaire, ideally it would be sought that they could place the information related to the criteria of validity and reliability of the evaluation. For example, the nine-item scale used to assess the presence or absence of stress symptoms should theoretically have information related to its validity and reliability. Likewise, it is important that the authors report information on the cut-off points they used for the classification of the anxiety / stress groups, in case there is no previous criterion, the authors could provide evidence that their cut-off points classify the examinees in different levels of anxiety / stress, this can be done in a simple way using an ANOVA test, or ideally through an unsupervised clustering method as k-means, where it is demonstrated that there are significant differences at every level of anxiety / stress.
The methods clearly explain that the authors use a bivariate technique (chi-square) to analyze the association between the attributes and each healthy behavior between the different anxiety / stress groups. Subsequently, reference is made to the logistic regression models with multiple predictors on a single dependent variable (the anxiety / stress groups).
Likewise, it is important to make an observation that could constitute a threat to the validity of the statistical conclusion, the chi-square test is sensitive to the sample size, so a small difference could be significant, it is also sensitive to the distribution between the categories, especially when there are cells with few values (less than 5 cases), this situation is observed in the “Not practiced-practiced” group, so it would be worthwhile for the authors to consider reformulating some analyzes as follows:
- Use a two-factor ANOVA (Health behaviors x anxiety / stress groups), using multiple comparisons (if the sample has homogeneous variances, then the Tukey-b and Scheffé tests should be used, in case the variances are heterogeneous the robust Games-Howell approach can be used). In any case, a parametric method is preferable because of the sample size.
- It is recommended that the authors consider the inclusion or not of the third group (Not practiced-practiced) since having cells with zero cases can affect the results of the statistical analysis. Maybe that group can be merged in order improve the model and results.
- It is important to consider that having an extreme group with few cases could potentially alter the significance of the results since the chi-square test only indicates if there are differences between the groups (regardless of where these differences are located).
- If after applying the parametric methods, the same conclusions are reached, the authors could assess the inclusion of these results instead of those obtained by non-parametric techniques.
Results and Discussion
Considering that similar results are achieved with the use of parametric techniques, I consider that the results are clearly presented, it would be worthwhile for the authors to place a little more interpretation of the findings after each table.
Particularly in the discussion it is suggested that the authors return to the findings obtained in the cited studies and compare the variables found in Japan with those found in other countries, even some of these differences could potentially be due to to the great cultural variations that exist between some countries. These types of discussions contribute to the objective of the manuscript, while others, such as the consequences of the decrease in visits to the dentist, are not necessarily related to this objective and rather address local issues.
Author Response
Thank you very much for reviewing our submission. Please see the attachment, and we have described the revisions to the paper with our comments.

Reviewer 2 Report
Overall, I found this a well written paper on an important topic in the covid era.
My comments for consideration by the authors are below.
- Lines 76-77: There was a poor response rate. Were follow-up reminders sent to help increase the response rate?
- Line 94: I recommend that table 1 is presented on the page after it is introduced in the body of the text.
- Lines 115-116: Since the participants were categorised into four groups, is there a reason why binomial rather than multinomial logistic regression was used?
- Lines 142-145: This text is the same as lines 132-135.
- Lines 169-170: Since tables should be ‘self-contained’ in journal articles, it would be useful to briefly describe ‘not practiced’, ‘practiced-not practiced’ , etc.
- Table 7: I am not familiar with the term ‘Ingenious mobility’. Is this the correct term?
- Table 7: I found this table difficult to understand. I can understand how dental visits would be defined, but how were the other outcome variables in this table defined?
Author Response

(The authors gave the same response as above.)
